# Developing and evaluating a lay health worker delivered implementation intervention to decrease engagement disparities in behavioural parent training: a mixed methods study protocol

Miya Barnett,[1] Jeanne Miranda,[2,3] Maryam Kia-Keating,[1] Lisa Saldana,[4] John Landsverk,[4] Anna S Lau[5]

For numbered affiliations see end of article.

**Correspondence to**
Dr Miya Barnett;
mbarnett@ucsb.edu

## ABSTRACT

**Introduction** Behavioural parent training (BPT) programmes are effective in preventing and treating early-onset conduct problems and child maltreatment. Unfortunately, pervasive mental health service disparities continue to limit access to and engagement in these interventions. Furthermore, challenges with parental engagement can impede the successful implementation of evidence-based practices (EBPs) in community settings that serve low-income, ethnic minority families. Lay health workers (LHWs)—individuals without formal mental health training—represent an important workforce to increase engagement, as they are members of the communities they serve. However, the mobilisation of LHWs has not been well studied as an implementation strategy to extend the reach or effectiveness of EBPs in the USA. LHW-delivered implementation interventions that specifically support the engagement of Latinx parents in evidence-based BPT programmes have the potential to improve clinical and implementation outcomes.

**Methods and analysis** A community-partnered approach will use the Quality Implementation Framework (QIF) to tailor and implement an LHW-delivered implementation intervention that aims to promote Latinx parent engagement in BPT programmes. Steps from the QIF will guide study activities to (1) conduct a mixed methods needs assessment to fit the implementation intervention to the local context, (2) adapt LHW-delivered implementation strategies to promote parent access to and engagement in Parent-Child Interaction Therapy and (3) conduct a hybrid effectiveness-implementation pilot trial to examine the feasibility, acceptability and preliminary effectiveness of the LHW implementation intervention at increasing engagement.

**Ethics and dissemination** Study procedures have been approved by the Institutional Review Board at the University of California, Santa Barbara. Results will be shared with the community-advisory group, at community-based meetings for other stakeholders involved in the pilot project, and submitted for publication in peer-reviewed journals.

### Strengths and limitations of this study

► This study seeks to develop and test an implementation intervention to address the impact of underutilisation and poor engagement in behavioural parent training (BPT) programmes, which limit their clinical effectiveness and successful implementation and sustainment.

► This study aims to improve mobilisation of lay health workers, who may be able to offer cultural and linguistic bridges to reach diverse families, as a potential solution to address racial/ethnic disparities in engagement in BPT programmes.

► As a pilot, this study is limited in its sample size to determine the effectiveness of the implementation intervention.

► This study will be limited in its generalisability due to the small sample size, the focus on one BPT programme (Parent-Child Interaction Therapy) and the characteristics of the local context.

## INTRODUCTION

Early-onset conduct problems and child maltreatment have been shown to have enormous personal and societal costs, including long-term mental health and substance abuse problems, higher service utilisation and future abuse against women and children.[1–3] Given that behavioural parent training (BPT) programmes have been shown to be effective at preventing and treating both child maltreatment[4] and conduct problems[5] for racially and ethnically diverse families,[6 7] large systems of care have invested millions of dollars in the implementation of these interventions.[8 9] Even with major implementation efforts, challenges remain with engagement and retention of families in BPT programmes.[10 11] A systematic review of engagement in BPT programmes found

that at least 25% of families that are appropriate for BPT programmes do not enrol in treatment, and an additional 26% begin, but then drop out of treatment, with higher rates of attrition for low-socioeconomic status families.[12] In fact, in community implementation of BPT programmes, attrition rates can exceed 65%.[13–15]

The consequences of poor participation in BPT programmes are significant. Families who drop out of treatment are less likely to experience improvements in parenting skills or child disruptive behaviours.[16] Moreover, failed efforts to recruit and retain parents are costly for providers.[17] Frequent cancellations and no-shows leads to fewer billable hours for community agencies, which are often under immense financial pressure.[17 18] Further, inadequate referrals negatively impact the implementation of evidence-based practice (EBP), as therapists may not learn to deliver the practice with fidelity.[8 9 19] Challenges with engagement may be especially pronounced for racial and ethnic minority families, as mental health service disparities have been well documented.[20] For example, African–American and Latinx children are almost 50% less likely than non-Latinx, white children to receive treatment for externalising disorders.[21]

In order to meet the public health potential of BPT programmes and address service disparities, implementation interventions are needed to support parental engagement for ethnic minority parents. Implementation interventions, which are usually complex and multilevel, include strategies to enhance the adoption and ongoing implementation of clinical interventions at the organisation, provider and consumer levels.[22] Multiple implementation strategies have been identified that focus on increasing consumer engagement with EBPs, including (1) increasing demand for EBPs, (2) intervening with consumers to enhance uptake and adherence and (3) preparing consumers to be active participants in treatment.[23] These implementation strategies are consistent with evidence-based approaches to improve engagement in children's mental healthcare, which include assessment of barriers, accessibility promotion, psychoeducation about services and appointment reminders.[24 25]

### Addressing mental health service disparities

Lay health workers (LHWs) may be especially well positioned to deliver consumer-facing implementation strategies focused on addressing service disparities for underserved, low-resource communities.[26] LHWs, which include a range of terms, including *promotores*, family peer advocates and wellness navigators, are individuals without formal mental health training, who have roles intended to increase their community's access to and benefit from services.[27 28] LHWs have the potential to address both demand and supply drivers of disparities in EBP delivery.[26] Demand for EBPs is impacted by an individual's mental health literacy, stigma towards mental illness and help seeking, perceptions of treatment providers and culturally based beliefs and preferences.[29 30] Systemic barriers to care may exacerbate disparities in accessing care. For example, undocumented immigrants are especially unlikely to seek mental health services due to fear of being reported to authorities.[31] Since LHWs come from similar cultural and personal backgrounds as the individuals they serve, they may be especially adept at helping patients overcome distrust of health systems.[32]

Regarding supply, the number of professional mental health providers who can deliver linguistically and culturally competent EBPs is inadequate.[33] The majority of mental health research with LHWs has been conducted in low-income and middle-income countries, with emerging evidence that LHWs can improve mental health outcomes when they are tasked with delivering EBPs.[28 34 35] Although LHWs have successfully delivered BPT programmes as prevention interventions in high-income countries, using a task-shifting model,[36–38] licensure and certification requirements frequently restrict EBP delivery to professionals in mental health settings.[26] Therefore, LHWs in the USA may need to have complementary and distinct roles within the provision of EBPs.[11 26] For example, if LHWs delivered auxiliary engagement services (eg, outreach and case management), it could reduce the burden on bilingual and bicultural mental health professionals and allow them to focus on activities that require advanced training and licensure, such as providing EBPs for more clients.[19 39]

One example of an LHW-delivered engagement programme is the Parent Empowerment Programme (PEP), which trains family peer advocates to work with parents to address their children's mental health needs and overcome barriers to care.[40 41] The majority of research on PEP has focused on evaluating the training of family peer advocates, as opposed to investigating the impact of the model on clinical outcomes for families, service utilisation or engagement in EBPs.[40–43] One randomised control trial evaluated the impact of PEP for black and Latinx parents of children with autism. Parents who received PEP had significantly lower stress than parents who received treatment as usual. However, there were no group differences for service utilisation. The researchers advocated that future research on programmes such as PEP should include non-English-speaking families, who may have higher levels of need, and use qualitative research to better understand the strengths and areas of improvement for the model.[44] The proposed study follows these recommendations through the development and evaluation of LHWs Enhancing Engagement for Parents (LEEP), an implementation intervention to improve engagement for low-income, Latinx parents into one BPT programme, Parent-Child Interaction Therapy (PCIT).[45] LEEP seeks to follow recommendations by Chacko and colleagues[12] based on their systematic review of parental engagement in BPT programmes by 'preparing parents for BPT, addressing practical barriers to engagement, assisting in aligning parent's involvement with their own goals for treatment' (p. 211) in order to impact initial and ongoing engagement in PCIT.

## Parent-child interaction therapy

PCIT has unique benefits and challenges related to engaging parents in treatment. The treatment model uses in vivo feedback to overcome challenges that are inherent to teaching methods used in other BPT programmes (eg, didactics, discussion) as it necessitates active participation and assesses learning in real time. PCIT requires that parents demonstrate a high level of proficiency with the targeted parenting skills before they advance from the first phase of treatment, which focuses on enhancing the parent-child relationship, to the second phase of treatment, which teaches effective and developmentally appropriate limit setting and discipline approaches, and then until they graduate from treatment.[45] Mastery-based criteria guarantee that all parents can successfully use the skills; however, parents often drop out before they learn the full range of parenting skills needed to decrease disruptive behaviours.[13 16] Furthermore, some research suggests that low-income, ethnic minority parents require more practice and time in treatment to reach this level of skill proficiency.[46–48] Extended treatment length can lead to long waitlists and fewer families seen in PCIT.[49] Further, problems with attendance, retention and prolonged skill acquisition have downstream effects on PCIT provider implementation outcomes. Clinicians can take up to 3 years to meet PCIT certification requirements (ie, achieving fidelity and graduating two cases).[50] Thus, low parental engagement results in provider attrition from training, which in turn compromises the sustainability of the intervention and limits the return on costly investments made to implement PCIT in public service systems.[8 9]

LEEP seeks to improve the supply of and demand for PCIT in agencies that predominately serve low-income, Latinx immigrant families, and address engagement challenges that impact clinical and implementation outcomes (figure 1). PCIT is widely implemented in community settings, including in the county where the current study is being conducted. LEEP includes LHW-delivered implementation strategies to increase caregiver engagement as an extension of PCIT services, which will be provided by the licensed mental health professionals. LEEP will be compared with PCIT implementation-as-usual to see if parental engagement and implementation challenges are ameliorated. A community-partnered approach will focus on making LEEP a feasible and acceptable implementation intervention, with the following aims.

### Aim 1

Assess the current context of LHW mobilisation in children's mental health services, to inform the development of LEEP.

### Aim 2

Through community partnership, develop a structure for the implementation of LEEP in publicly funded, children's mental health settings.

### Aim 3

Evaluate the feasibility of implementing LEEP in community mental health agencies through a pilot effectiveness-implementation trial.

## METHODS AND ANALYSIS

### Conceptual framework and approach

The Quality Implementation Framework (QIF), which includes four phases to support high quality implementation, informs the study aims and plans for scaling-up LEEP (figure 2).[51] The first phase of the QIF focuses on assessing organisational needs, readiness and innovation-organisational fit, which will be conducted in aim 1 through survey data and stakeholder interviews. Phase 2 in the QIF focuses on the development of implementation structures, which will occur during the second aim with the input of a community-advisory group. The community-advisory group will collaboratively help to develop an implementation plan

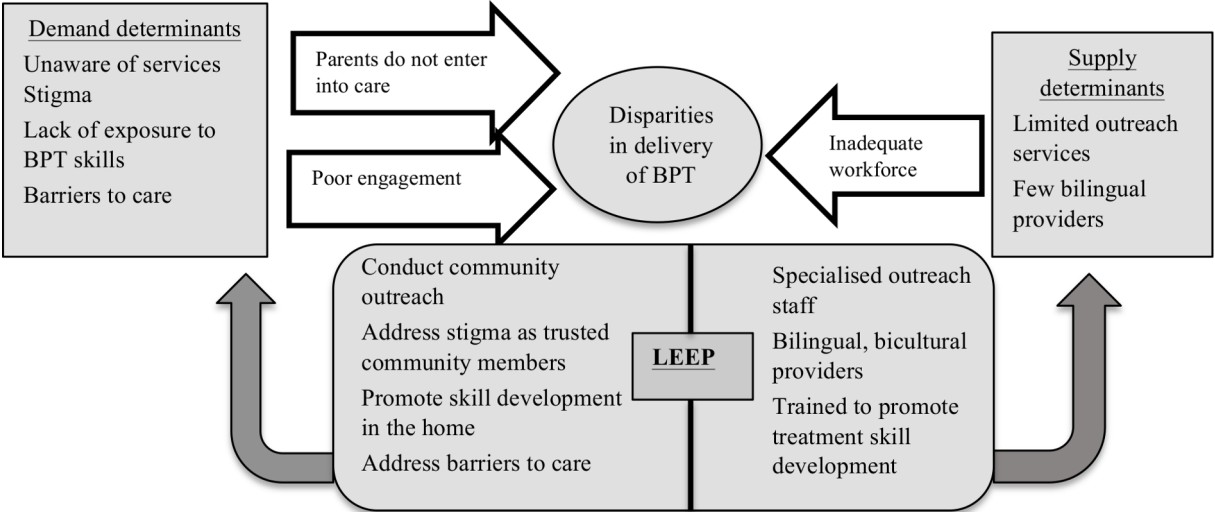

**Figure 1** LEEP's approach to address supply and demand determinants of disparities. Adapted from Barnett et al.[26] BPT, behavioural parent training programme; LEEP, LHW Enhancing Engagement for Parents; LHW, lay health workers.

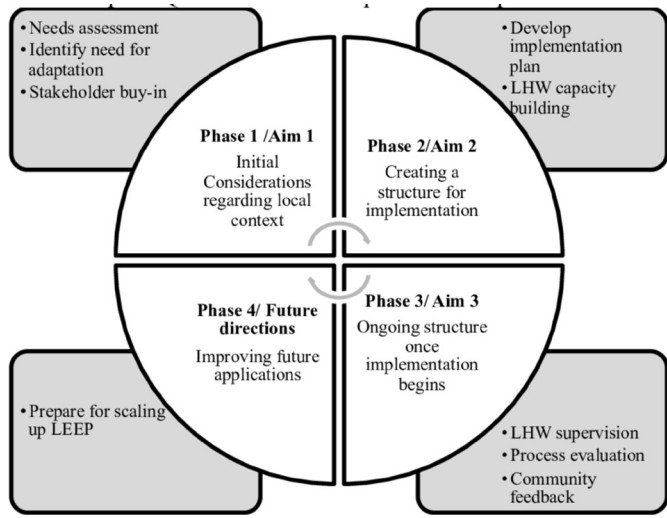

**Figure 2** Critical steps of QIF in LEEP development and implementation. Adapted from Meyers *et al.*[51] LEEP, LHWs Enhancing Engagement for Parents; LHW, lay health worker.

delineating tasks and timelines to establish infrastructure for LHW capacity building, including job descriptions and training plans. Phase 3 of the QIF includes three main activities that will take place during a hybrid type 2 effectiveness-implementation pilot stepped-wedge trial of LEEP. These activities include (1) providing implementation support strategies (eg, supervision and consultation) to LHWs, (2) conducting a process evaluation to identify successes and barriers to implementation and (3) providing ongoing feedback to organisations about the impact LEEP is having on service outcomes. Finally, aim 3 activities will inform phase 4 of the QIF, which focuses on learning from initial implementation experiences to inform future efforts to scale-up and sustain LEEP.

### Patient and public involvement

The research questions, study design and outcomes measures were informed by public stakeholders, including agency leaders, PCIT therapists and LHWs. No patients were involved in this process, although LHWs often have shared characteristics and life experiences given their social proximity to the individuals in the communities they serve.[52] The burden of the randomised control trial will be evaluated through the collection of feasibility and acceptability data in qualitative interviews with participants. Results will be shared with participants in community-based events.

### Aim 1

Assess the current context of LHW mobilisation in children's mental health services, to inform the development of LEEP.

### Participants

Surveys will be administered to LHWs employed or contracted by children's mental health agencies in two counties in California. Approximately 70 LHWs will be recruited to complete the quantitative survey. Based on a national survey of LHW,[53] LHWs are expected to be Latinx, female and have below a college level of education. Ten agency community mental health agency leaders and 25–30 LHWs will be invited to participate in interviews to expand on the findings from the survey. Survey and interviews will be offered in Spanish or English.

### Procedure

A mixed method needs assessment will be conducted to understand how LHWs are currently mobilised in children's community mental health settings, with the purpose of adapting LEEP to fit within the local context. Surveys will provide a breadth of information and qualitative interviews will provide depth of information, to understand perceived barriers to parental engagement in children's mental health services, LHW roles and integration into services, and LHW knowledge about and attitudes towards BPT programmes and evidence-based engagement strategies. Data collection started in January 2017 and was completed in December 2018.

### Survey measures

Surveys will be collected via electronic or paper-and-pencil survey based on LHW preferences.

**LHW characteristics**[39] A demographic questionnaire will provide information about the LHWs' characteristics, including gender, race/ethnicity, country of origin, educational level and years of experience.

**Cultural Background Questionnaire**[54] The Cultural Background Questionnaire is a 19-item self-report measure used to assess therapist generational status and acculturation, including cultural identity (ie, US identity and Heritage Cultural Identity) and language use.

**Parental Engagement**[55 56] A questionnaire that was developed to measure provider's perceptions of and strategies for engaging fathers has been adapted to measure LHW's perceptions of barriers to engagement, strategies for engagement and confidence in engagement for parents. The adapted questionnaire includes perceived barriers in engagement for parents in general, and the LHW's use of and confidence with engagement strategies for both mothers and fathers.

**Attitudes towards BPT strategies.** A four-item questionnaire was developed for this study to measure LHW's attitudes towards teaching parents common strategies targeted in BPT programmes including play to improve the parent-child relationship, praise of positive behaviours, ignoring minor misbehaviours and time-out as a form of discipline.

**EBP Questionnaire**[57] A questionnaire developed to measure service broker's knowledge of and referrals to EBPs has been adapted to identify if LHWs are aware of, making referrals to and supporting families involved specifically in PCIT (eg, *'Have you referred parents to Parent-Child Interaction Therapy (PCIT)'*).

### Semistructured Interviews

Interview guides will include topics, questions and probes related to LHW roles, training needs and experiences and

attitudes related to BPT programmes. Questions will investigate how LHWs view their roles in agencies and their communities, their perceptions of BPT programmes, their preparation for their position and their training needs. Interviews with agency leaders will focus on how LHWs are integrated into services, LHW training, and outreach and engagement strategies with Latinx families. A 'funnel-approach' will be used with broad open-ended questions related to roles, trainings and attitudes asked first, followed with specific probes to elicit details.[58]

### Analysis

A QUAN+QUAL mixed methods design will be used, with quantitative and qualitative data collected simultaneously and given equal weight in analyses, for the purposes of gaining breadth and depth of understanding (ie, complimentarity), identifying if the qualitative and quantitative data provide the same answer to the same question (ie, convergence), and using qualitative data expand on unexpected quantitative findings.[59] Interviews will be transcribed and entered into NVivo, a software that aids the coding, organisation and retrieval of codes. An iterative process will be used where the coding team first develops a preliminary coding scheme and applies it to a sample text to ensure all relevant themes are captured. Once a final coding scheme is decided on, coders will apply the final code list to all transcripts. Regular meetings with the coding team will be conducted to examine coding across analysts, resolve differences in coding, conduct iterative refinement of code definitions and the logic of the coding tree, and collaborate on the development of themes. Qualitative themes will be identified through analysis of co-occurring codes and text analysis.[60 61]

### Aim 2

Through community partnership, develop a structure for the implementation of LEEP in children's mental health settings.

### Participants

A community-advisory group with six to nine stakeholders will be formed to make sure that implementation supports match the local context. Agency leaders, PCIT therapists and LHWs will be represented in the advisory group. Given the wide diversity of viewpoints, education levels and ethnicities, efforts will be made to provide each participant with equal representation, opportunities for contribution and honorariums.

### Procedure

In line with the Model of Research-Community Partnership, which was specifically developed for research in children's mental health services, the formation of the partnership will focus on building relationships, trust, establishing a joint mission and identifying roles and responsibilities of different partner members. This will provide the foundation to build a synergistic, collaborative relationship focused on developing and delivering LEEP, which in turn could improve the successful and sustained implementation of PCIT.[62] Using data from aim 1 and in collaboration with the community-advisory group, the LEEP implementation intervention will be adapted from an existing protocol focused on LHW-delivered parent outreach and engagement strategies. This protocol was developed to increase access to PCIT in a low-income, Latinx community in the Southeastern USA, but has not been disseminated to other communities.[39] The implementation supports needed for LHWs to deliver LEEP also will be identified and put into place. Steps from the QIF will be used to guide the activities of the community-advisory group in adapting these materials and developing LEEP's implementation structure.[51] Advisory group meetings will include (1) activities to build trust and develop a shared mission statement, (2) feedback on adapting LEEP materials, (3) advisory group input on survey and interview results from aim 1 and (4) development of a plan with specific tasks, roles, tracking for LEEP implementation. Phase 2 activities involving the community-advisory group began in December 2018 and will continue through March 2021.

### Aim 3

Evaluate the feasibility of implementing LEEP through a pilot effectiveness-implementation trial.

In aim 3, an effectiveness/implementation hybrid design (type 2) pilot study will integrate qualitative and quantitative data to examine the feasibility of delivering and scaling-up LEEP. Type 2 hybrid trials simultaneously measure the clinical effectiveness of an intervention, in this case PCIT, and the feasibility and utility of an implementation intervention (ie, LEEP).[63] Pilot studies are limited in their ability to test effectiveness given small sample sizes, but they provide a critical phase of research design that can examine the feasibility of the approach to be used in a large-scale study.[64] A focus will be placed on measuring engagement outcomes at a client and agency levels to evaluate if LEEP is increasing the reach of PCIT services, service entry and treatment engagement.

### Procedure

Three agency sites that provide PCIT will be involved in this pilot study, which will use a stepped-wedge design. In a stepped-wedge design, a period of baseline measurement will occur for all sites, in which PCIT will be implemented-as-usual. Then at subsequent time points, each site will be randomised to LEEP and response to the intervention will be measured for client and implementation outcomes (figure 3). At each agency, one to two LHWs (four to six in total) will be trained to deliver LEEP. Client and implementation outcomes will be collected during PCIT implementation-as-usual and LEEP implementation. The baseline measurement period began in January 2019. LHWs will be trained to deliver LEEP at the first site starting in July 2019 and will continue through March 2021.

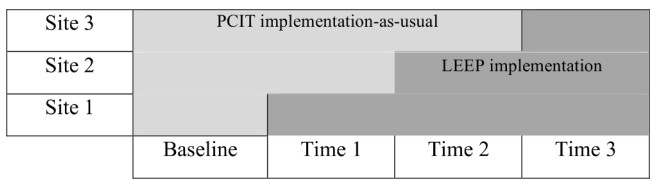

**Figure 3** Stepped-wedge trial of LEEP implementation. A stepped-wedge design will be used with the three sites implementing LEEP at separate time points. LEEP, LHWs Enhancing Engagement for Parents; LHW, lay health worker; PCIT, Parent-Child Interaction Therapy.

### Participants

Four to six LHWs will be trained to provide LEEP. These LHWs will provide LEEP care extension services to approximately four PCIT clients each (16–24 families). LHWs will be trained to provide informed consent for parents. Families that participate in LEEP or PCIT implementation-as-usual will meet criteria for receiving this BPT programme. This includes having a child between the ages of 2 and 7 and presenting problems consistent with disruptive behaviours, or risk for child maltreatment.

### LEEP intervention

Based on community-advisory group feedback on the needs of the Latinx immigrant community and the agencies implementing PCIT, along with research on parental engagement,[11] LEEP includes components for LHWs to (1) increase awareness of PCIT for Latinx immigrant families, (2) promote engagement once parents seek PCIT services and (3) support parents' use of skills taught in PCIT throughout treatment. To increase knowledge of PCIT, LHWs will conduct community presentations in locations with parents of young children (eg, Head Start Centres, churches). Parents will be referred to LEEP when they first seek services to promote engagement. LHWs will meet with parents in their home to discuss identify how PCIT aligns with their goals for treatment, address practical barriers to engagement (eg, transportation) and introduce the relationship-enhancing parenting skills taught in PCIT. Once parents start PCIT, LHWs will provide home visits to promote skill practice and treatment adherence at the beginning of each treatment phase. Additional booster sessions will be provided based on the parent's progress in treatment. If parents have not reached mastery criteria after five sessions, LHWs will conduct weekly home visits to reinforce skill use and address barriers to engagement. LHWs will be provided with electronic tablets with e-books that included scripts and videos to use in one-on-one meetings with parents before they enter into care and while they are receiving PCIT services. The e-books will have materials to help LHWs promote motivation (eg, parent testimonials), homework adherence and skill practice (eg, video demonstrations of the targeted parenting skills).

### Effectiveness outcome measures

Given that PCIT is an assessment-driven BPT programme, clinical outcomes will be assessed using standard measures

that are collected as part of the routine PCIT protocol. Parents will not complete any additional measures for this study.

*Engagement.* To assess if LEEP impacts engagement at the family level, session attendance, graduation from PCIT and the number of sessions needed to graduate will be assessed. Further, daily skill practice will be measured using the record sheets that parents complete over the week, which has been used in the past studies on homework adherence.[65]

*The Dyadic Parent-Child Interaction Coding System (DPICS).*[66] The DPICS is a behavioural observation coding system that was designed to measure the quality of interaction in parent-child dyads, which has good inter-rater reliability. This study will use the DPICS categories, *Behaviour Description, Labelled Praise, Unlabelled Praise, Reflection, Question, Negative Talk, and Indirect and Direct Commands,* to measure the parent's skill acquisition.

*Eyberg Child Behaviour Inventory (ECBI).*[67] The ECBI is a 36-item parent-rating scale of disruptive behaviour problems for children between the ages of 2 and 16. Parents rate the frequency of each disruptive behaviour on a seven-point Likert scale ranging from *never* (1) to *always* (7), which are summed to yield the Intensity Scale and whether this behaviour is a problem for them, with the total number of yes responses yielding the Problem Scale.

### Implementation outcomes

Using the implementation outcomes outlined by Proctor and colleagues,[68] this study uses mixed methods to understand the acceptability, appropriateness, feasibility, reach and costs of delivering LEEP.

### LHW level outcomes

To measure changes in LHW knowledge, perceptions of acceptability and feasibility of PCIT, and competence, LHW will complete pre-self-report and post-self-report and behavioural measures (table 1). Ongoing fidelity monitoring will be conducted throughout LHW's delivery of LEEP. Fidelity monitoring will include reviewing data capture of the ebook created for LEEP, which will include videos about PCIT and scripts for the LHWs to use with the families they serve. LHWs will also audio record their sessions to monitor fidelity to the LEEP model.

### Implementation costs

Costs associated with delivering LEEP will be measured by calculating time estimates associated with all aspects of implementation.

### Agency efficiencies

To identify if LEEP impacts agency efficiencies, with therapists increasing their billable hours, administrative claims will be calculated for PCIT therapists to measure the time spent in direct services.

### Reach and penetration

At the agency level, reach of PCIT will be assessed by the number of clients that enrol in and graduate from PCIT.

**Table 1** Measures of LHW training and outcomes

| Measure | Description | Administration |
|---|---|---|
| Demographic and Background Form[39 54] | Characterises personal and professional backgrounds of LHW | Aim 1: survey<br>Aim 3: training |
| EBP Questionnaire[57] | Measures service brokers' awareness of and referrals to EBPs. | Aim 1: survey |
| Parent Engagement Strategy Use and Confidence[55 56] | Assess use and confidence of engagement strategies with mothers and fathers. | Aim 1: survey<br>Aim 3: pretraining, post-training |
| Acceptability and Feasibility of PCIT[69] | Assess LHW perceptions of the acceptability and feasibility of PCIT | Aim 1: survey<br>Aim 3: pretraining, post-training |
| PCIT Knowledge Quiz[39] | A quiz that measures the knowledge of PCIT principles and practices | Aim 3: pretest, post-training |
| Dyadic Parent-Child Interaction Coding System[66] | A behavioural observation coding system that assesses parent-child interactions. It will be used to measure LHW's ability to model parenting behaviours. | Aim 3: pretraining, post-training |
| LEEP Fidelity Monitoring | Review of ebook data capture to see the LEEP resources being used and audio recordings of home sessions. | Aim 3: during consultation |

EBP, evidence-based practice; LEEP, LHWs Enhancing Engagement for Parents; LHW, lay health worker; PCIT, Parent-Child Interaction Therapy.

Using administrative claims data, penetration at the agency level will be calculated as the percentage of children receiving PCIT in out of the number of children who are eligible for this EBP. Furthermore, the percentage of families that successfully complete PCIT out of the families enrolled will be calculated.

### Acceptability and feasibility

Qualitative interviews will be conducted with the LHWs, agency leaders and 10 parents to assess their perceptions of LEEP including perceived acceptability, appropriateness and feasibility, which are important early implementation outcomes.[68]

### Analysis

This pilot trial is designed to evaluate the feasibility of implementing LEEP and develop tools to measure its clinical and implementation targets and outcomes. The trial is not powered to assess intervention effects. Analyses will focus on establishing the reliability and validity of measures of clinical engagement and implementation outcomes. Qualitative data will be analysed using the methodology described in aim 1. Qualitative and quantitative data will be given equal weight in analyses with a focus on *convergence, expansion* and *complimentarity,* with quantitative data used to measure outcomes and qualitative data to understand process.[59]

### DISCUSSION

LEEP has the potential for a significant public health impact, by developing an implementation intervention to increase entry and engagement of Latinx parents into BPT programmes to improve clinical and implementation outcomes. Although LHWs have been identified as an important workforce to address mental health disparities, limited research has evaluated the best strategies to mobilise them to support EBP implementation in the USA.[28] As a pilot study, findings will be limited in power and generalisability. However, the exploratory and development work in this study will provide data on the feasibility and acceptability of LEEP and its preliminary impact on client recruitment, adherence and retention in PCIT, which will inform future scaling-up of the model.

### Ethics and dissemination

Study procedures have been approved by the Institutional Review Board at the University of California, Santa Barbara. Results will be submitted for publication in peer-reviewed journals. Furthermore, results will be shared with the community-advisory board and other stakeholders involved in the pilot of LEEP.

**Author affiliations**
[1]Counseling, Clinical, and School Psychology, University of California Santa Barbara, Santa Barbara, California, USA
[2]Psychiatry and Biobehavioral Sciences, University of California Los Angeles, Los Angeles, California, USA
[3]Center for Health Services and Society, University of California Los Angeles, Los Angeles, California, USA
[4]Oregon Social Learning Center, Eugene, Oregon, USA
[5]Psychology, University of California Los Angeles, Los Angeles, California, USA

**Acknowledgements** The authors would like to thank Camille Nebeker and Emily Winslow, who serve asconsultants on this project. We are grateful to the help provided by JuanCarlos Gonzalez, Corinna Klein, Erika Luis Sanchez, Iliana Flores, and AnaRomero, who assisted on this project. Finally, we would like to thank ourcommunity partners at the Santa Barbara Promotora Network and CALM (Child AbuseMediation and Listening), without whom this research would not be possible.

**Contributors** MB is the principal investigator for the study protocol. MB generated the idea and design of the study, and was the primary writer of the manuscript. ASL, JM, JL, MKK and LS also made substantial contributions to study conception and design as mentors to MB for this K01 award. All authors reviewed and provided feedback for this manuscript. The final version of this manuscript was vetted and approved by all authors.

**Funding** This study was funded by the National Institute of Mental Health (K01MH110608) awarded to MB. Additionally, the preparation of this article was supported in part by the Implementation Research Institute (IRI) at the Brown School, Washington University in St. Louis, through an award from the National Institute of Mental Health (R25 MH080916). MB is a fellow of IRI, ASL and LS are past fellows of IRI, and John Landsverk is senior faculty of IRI.

**Competing interests** None declared.

**Patient consent for publication** Not required.

**Ethics approval** University of California, Santa Barbara IRB approved study procedures (Protocol number: 1-18-0919 and 4-19-0167).

**Provenance and peer review** Not commissioned; externally peer reviewed.

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
