## [Reviewer comments · BMJ Open]

ARTICLE DETAILS

TITLE (PROVISIONAL)	Developing and Evaluating a Lay Health Worker Delivered Implementation Intervention to Decrease Engagement Disparities in Behavioral Parent Training: A Mixed Methods Study Protocol
AUTHORS	Barnett, Miya; Miranda, Jeanne; Kia-Keating, Maryam; Saldana, Lisa; Landsverk, John; Lau, Anna

VERSION 1 - REVIEW

REVIEWER	Terje Ogden Norwegian Center for Child Behavioral Development, Norway
REVIEW RETURNED	28-Jan-2019

GENERAL COMMENTS	This is a very promising study with a well structured and thorough design. The outcomes of the study will undoubtedly have the potential to improve practice in the process of engaging low income and ethnic minority families in the PCIT or other behavioral parent training programs. The introduction gives a broad and relevant presentation and discussion of barriers to parent engagement in BPTs. I would, however, recommend the authors to be more explicit about previous research on the impact of LHW in mental health, and present a more coherent summary of research. It is claimed that there is limited research on 1) whether and how the consumer-facing strategies might increase the success of EBP implementation (p. 5) and 2) how LHW-delivered strategies impact implementation and clinical outcomes of EBPs and if they successfully reduce disparities in engagement (p.6). It is unclear from these formulations whether such research exists, and if it does I would encourage the authors to refer and comment on these studies. Given the aims of the study, it has a very relevant and strong mixed method design which is implemented in a stepped wedge fashion. The implementation study is planned according to the Quality Implementation Framework and the well-established Parent-Child Interaction Therapy program. Being a pilot feasibility study, it is not powered to be a randomized trial. Moreover, the authors give a very clear presentation of the strengths and limitations of the study (p. 3) and this presentation is also nicely elaborated on later in the manuscript. The study protocol is better developed when it comes to the research implementation and the PCIT, than on the implementation of LHW practice. It is mentioned (p.13) that the LEEP implementation intervention will be adapted from a protocol
---

	focused on LHW-delivered parent outreach and engagement strategies. It is also mentioned that the LHWs will conduct community presentations and that they will be supplied with electronic tablets to use in one-to-one meetings with parents (p. 15). But I find this to be an insufficient description of the intervention and recommend the authors to be more explicit about the practical components involved. It would be interesting to know more about the practical framework for how the LHWs are planned to contact and engage families (e.g. how many visits are intended, where, and for how long will each visit last, for how long will the LHWs work with the families, and will they stay in contact with the families once they have entered treatment?)
--	---

REVIEWER	Anil Chacko New York University USA
REVIEW RETURNED	10-Feb-2019

GENERAL COMMENTS	A Mixed Methods Study to Develop and Evaluate a Lay Health Worker Delivered Implementation Intervention to Decrease Engagement Disparities in Behavioral Parent Training This is an important study focused on increasing access and engagement to behavioral parent training through lay health workers. I provide a few recommendations that I hope will enhance the clarity of this manuscript. 1. The manuscript would benefit from a review of more recent empirical data on engagement and retention of families and behavioral parent training. Chacko and colleagues published a systematic review of that literature in 2016 which provides data on engagement at various stages within behavioral parent training, which would be a useful and more up-to-date comparator for the purposes of engagement outcomes in the study Chacko, A., Jensen, S., Lowry, L.S., Cornwell, M., Chimiklis, A., Chan, E. , Lee, D., & Pulgarin, B. (2016). Engagement in Behavioral Parent Training: Review of the literature and implications for practice. Clinical Child and Family Psychology Review, 19, 204-215. 2) There has been a increase in the use of lay health workers in the delivery of evidence-based practices, including BPT. I suggest the author's review some of these studies noted below, which provide some useful context on how non and paraprofessionals, including LHWs, have been utilized and approaches to engagement and retention in behavioral parent training: Chacko, A., Fabiano, G., Doctoroff, G. & Fortson, B. (2018). Engaging fathers in effective parenting for preschool children using shared book reading: A randomized controlled trial. Journal of Clinical Child and Adolescent Psychology, 47, 79-93. Chacko, A., & Scavenius, C. (2017). Bending the curve: Community based behavioral parent training to address ADHD symptoms in the voluntary sector in Denmark. Journal of Abnormal Child Psychology
---

Chacko, A., Gopalan, G., Franco, L., Dean-Assael, K., Jackson, J., Marcus, S., Hoagwood, K., & McKay, M. (2015). Multiple-Family Group service delivery model for the children with disruptive behavior disorders. *Journal of Emotional and Behavioral Disorders*, 23, 67-77.

3) It Seems that the use of PCIT Is because the community settings already use this intervention approach; is that accurate? If this is a decision point it's unclear to me why pcit should be utilized, given the state of the literature comparing this approach to other BPT approaches. As an example, there's been some recent work comparing group based behavioral parent training to pcit conducted by Deborah Gross and colleagues. The findings suggest equivalent benefits of these two approaches, but significant reduction in cost and increased access to treatment in favor of the group based BPT model. The decision to use PCIT is an interesting one and arguably poses more challenges.

While the LEEP approach is aimed toward increasing engagement and adherence to PCIT, given that work is done in community-based mental health agencies, how will LEEP improve access to services? It's unclear to me how a setting that has already long waiting lists and difficulties accessing services will be improved simply by having lay health workers implement the LEEP approach? Aren't these settings already at full capacity for provision of Mental Health Services? Are lay health workers and additional work force that can provide additional mental-health services? This is unclear in the manuscript.

4) It will be helpful to know at what stage this research is in and if it has already gone through several of these aims, what has been learned thus far.

5) It would be helpful to have a more clear description of what the intervention is. PCIT is The intervention approach which is combined with LEEP, which is an implementation approach. I assume here that mental health professionals will be providing PCIT and lay health workers will be providing LEEP? At various points in the manuscript, it read like both PCIT and LEEP would be delivered by lay health workers. This needs to be made much more clear and stated early in the manuscript.

6) Given the commonalities between LEEP and what New York State utilizes in the Parent Empowerment Program (Hoagwood and Jensen), it would be helpful for the authors to review that work. It's likely that many of the strategies to improve barriers to care will be similar.

Rodriguez, J., Olin, S. S., Hoagwood, K. E., Shen, S., Burton, G., Radigan, M., & Jensen, P. S. (2011). The development and evaluation of a parent empowerment program for family peer advocates. *Journal of Child and Family Studies*, 20(4), 397-405.

7) How will the measure of "the percentage of days that daily skill was completed" be assessed (page 15 line 47)?

8) An understudied phenomenon that I think it's important moving forward in the literature on behavioral parent training is adverse events related to the delivery of behavioral parent training. It may

	be helpful for the authors to consider assessing this data in their study. See Allan and Chacko 2018 for discussion of this in the context of behavioral parent training for ADHD. Allan, C. & Chacko, A. (2018). Adverse Events in Behavioral Parent Training: An Under-Appreciated Phenomena. The ADHD Report
--	---

VERSION 1 – AUTHOR RESPONSE

Reviewer Comments:

5. Reviewer 1 and 2 requested more discussion of the research that exists around LHW-delivered strategies to improve EBP implementation. Specifically, Reviewer 2 made several excellent recommendations on the existing literature of task-shifting (Chacko et al., 2015; Chacko et al., 2018; Chacko & Scavenius, 2017) within BPTs and the relevance of Hoagwood and colleagues' work with the Parent Empowerment Program.

We thank the reviewers for their excellent recommendations to increase our review of the current literature related to LHW-delivered strategies to improve EBP implementation. We have expanded our review on implementation efforts with LHW in the following ways: First, we have included the recommended references related to how task-shifting has been used in BPT delivery, especially in prevention programs. Second, we now have a paragraph on pages 6-7 that highlights the Parent Empowerment Program (PEP). We have described the program and highlighted the evidence on it to date. Finally, we identify the current gaps in the literature, regarding how PEP has not specifically been tied to the implementation of EBPs.

6. Reviewer 1 and 2 requested more information about LEEP, including the roles of LHWs as compared to PCIT therapists and the length and frequency of LEEP sessions.

We agree that the description of the LEEP intervention was limited previously, and now has been greatly expanded on page 15, within the Method section. Further, in response to clarification requested by Reviewer 2, we have explained what components the LHWs are delivering and which components the PCIT therapists are providing on page 8 in the introduction, stating, "LEEP will include LHW-delivered implementation strategies to increase caregiver engagement as an extension of PCIT services, which will be provided by the licensed mental health professionals."

7. Reviewer 2 recommended that the manuscript would benefit from a review of more recent empirical data on engagement and retention of families and behavioral parent training.

We appreciate the recommendation of the review from Chacko and colleagues (2016), Engagement in Behavioral Parent Training: Review of the literature and implications for practice. We have cited this work throughout the paper. Specifically, we present results from the review on page 4, "A systematic review of engagement in BPTs found that at least 25% of families that are appropriate for BPTs do not enroll in treatment, and an additional 26% begin, but then drop out of treatment, with higher rates

of attrition for low-SES families.¹¹ In fact, in community implementation of BPTs, attrition rates can exceed 65%.^{11–13} We also show how LEEP aligns with recommendations from this review on page 7, stating, “LEEP seeks to follow recommendations by Chacko and colleagues based on their systematic review of parental engagement in BPTs, by focusing on, “preparing parents for BPT, addressing practical barriers to engagement, assisting in aligning parent’s involvement with their own goals for treatment” (p. 211) in order to impact initial and ongoing engagement in PCIT.”

8. Reviewer 2 identified that it was not clear why PCIT was selected as the intervention. Notably, Reviewer 2 acknowledged that PCIT has unique challenges. Specifically, they requested further information to address how LEEP will improve access to services given that community mental health agencies are likely to have long waitlists and be at full capacity for provision of Mental Health Services.

We appreciate the concerns that Reviewer 2 has identified regarding PCIT. We have highlighted that this intervention was selected as it has been widely implemented in community-settings, including the county where the study is being conducted. Challenges with PCIT implementation from the Gross et al. (2018) study, recommend by Reviewer 2, are highlighted as needing to be addressed, including how long-waitlists for this treatment occur when families do not progress in treatment. For example, in the Gross et al. (2018) study, families were in treatment for over a year on average, even though PCIT is intended to be a shorter treatment. LEEP intends to address challenges with PCIT implementation specifically, by targeting strategies to enhance the efficiency of skill acquisition and engagement throughout the treatment protocol, which could increase the number of families who are served and successful in this EBP.

9. Reviewer 2 to stated that it would be helpful to know at what stage this research is in and if it has already gone through several of these aims, what has been learned thus far.

We have included the dates of each aim of the study. Currently, we are completing community-partnered activities in Aim 2. We have highlighted how the partnership procedure has informed the elements of LEEP, including the points at which the LHWs will contact parents and the procedure to do so. To maintain the scope of a protocol paper, further results from the research are not presented.

10. Reviewer 2 requested information on, how the percentage of days that daily skill was completed will be assessed.

On page 16, we now include, “daily skill practice will be measured using the record sheets that parents complete over the week, which has been used in past studies on homework adherence.”

11. Reviewer 2 recommended that we assess data to evaluate adverse events related to the delivery of behavioral parent training and recommended the article, Allan, C. & Chacko, A. (2018). Adverse Events in Behavioral Parent Training: An UnderAppreciated Phenomena. The ADHD Report.

We appreciate this excellent recommendation and agree it is important to identify adverse events related to the delivery of behavioral parent training programs. However, given the current scope of the project on community-delivered PCIT, we are not able to expand our focus to include this data at this time.

VERSION 2 – REVIEW

REVIEWER	Terje Ogden Norwegian Center for Child Behavioral Development
REVIEW RETURNED	08-Apr-2019

GENERAL COMMENTS	The Authors have been very receptive to the reviewers' comments, and I recommend this manuscript for publication
--

REVIEWER	Anil Chacko New York University, USA
REVIEW RETURNED	09-Apr-2019

GENERAL COMMENTS	I thank the authors for their efforts on addressing the comments I had on the previous manuscript. The authors have done an excellent job on the revisions and I have no further comments.
--